# LIGHT-IMPLICIT UNCALIBRATED PHOTOMETRIC STEREO NETWORK WITH FOURIER EMBEDDING

## ABSTRACT

We present a one-stage deep uncalibrated photometric stereo (UPS) network for non-Lambertian objects. Previous two-stage deep UPS networks estimated surface normals based on learned lighting because lighting is tangled with shading cues, making it challenging to directly estimate surface normals. However, two-stage UPS networks face fewer interpretations with embedded light direction's role in decomposing shading cues. Additionally, these two-stage methods discretize the light direction estimations instead of regressing exact light directions due to the learning difficulty and instability. However, the inexact light directions mislead shading cues extracted by the normal estimation network.

In contrast to previous two-stage UPS methods, our UPS-FourNet implicitly learns lighting by decomposing inputs using embedded Fourier transform. Our approach is motivated by a unique observation from photometric stereo images in the Fourier domain: lighting information predominantly concentrates on phases while shape information is closely related to amplitudes. By leveraging this property, the shape and lighting can be "decomposed" to a certain extent in the Fourier domain, eliminating the need for explicitly learning light directions and using them in the subsequent normal regression network. UPS-FourNet relaxes the limitations of two-stage UPS methods, with better training stability, concise end-to-end structures, and avoiding the discrete classification errors of light directions. Experiments on synthetic and real datasets show that our method achieves competitive results, which may push a new strategy for deep learning-based UPS methods.

## 1 INTRODUCTION

Photometric stereo (PS) aims at recovering the surface normal of an object from various shading cues under multiple images with different lights Woodham (1980). Compared with geometric stereo methods, photometric stereo methods are excellent at capturing high-frequency details on objects' surfaces. Therefore, PS plays a mainstream role in the fine-detailed surface recovery needed in scientific and engineering areas such as cultural relics digitizationZhou et al. (2013), forensics Sakarya et al. (2008), and industrial detection Ren et al. (2018).

Most of the existing PS methods, *i.e.*, calibrated photometric stereo (CPS) Chandraker et al. (2012); Chen et al. (2017), require knowledge of the light directions for each image. However, calibrating the light directions involves complex operations and relies on specialized instruments, making it impractical for real-world applications. Conversely, uncalibrated photometric stereo (UPS) Papadhimitri & Favaro (2013); Lu et al. (2015) can estimate surface normals without lighting information. However, UPS faces more challenges than CPS, such as the Generalized Bas-Relif (GBR) ambiguity Belhumeur et al. (1999), which is the inherent inability to uniquely determine the shape and reflectance of a surface solely from the observed image intensities due to the lack of light source directions. On the other hand, UPS also encounters the challenge of general non-Lambertian surface reflectance in the real world. More badly, resolving the GBR ambiguity often requires assuming a simplified Lambertian reflectance model Shi et al. (2010); Papadhimitri & Favaro (2014). Although some methods Lu et al. (2013; 2015) can handle surfaces with general bidirectional reflectance distribution functions (BRDFs), they are restricted to a uniform distribution of light directions.

Recently, deep learning-based PS methods have demonstrated impressive results in dealing with general reflectance and complex structures, owing to the powerful capabilities of deep neural net-

works Chen et al. (2018); Ikehata (2018); Li et al. (2019); Chen et al. (2020a); Honzátko et al. (2021); Ju et al. (2022b). Nevertheless, there has been relatively limited focus on deep learning-based UPS methods. Among existing approaches, the CPS method PS-FCN Chen et al. (2018), can address the UPS problem by directly learning the mapping from input images to surface normals without concatenating light directions, denoted as UPS-FCN. However, the performance of UPS-FCN is far behind satisfactory due to the complex coupled among shading cues, encompassing unknown lighting directions, surface normals, and reflectance properties. Therefore, all subsequent deep learning-based UPS methods Chen et al. (2019; 2020b); Sarno et al. (2022); Li & Li (2022b) also adopt a two-stage strategy, that is, first estimating the light directions and then estimating the surface normals using both the estimated light information and input images.

However, the two-stage network strategy also brings certain challenges. First, the existing methods Chen et al. (2019; 2020b); Sarno et al. (2022) concatenate the expanded light directions with the input images and use CNN-based encoders to approximately decouple the features of surface normals. Although this approach has achieved good results, it lacks a clear physical interpretation. The role of embedded light direction in decomposing shading cues in neural networks remains less intuitive for researchers to comprehend and improve upon the results. Also, two-stage methods suffer from training instability and tedious non-end-to-end training way. On the other hand, previous deep UPS methods Chen et al. (2019; 2020b); Sarno et al. (2022) have to convert the light direction estimation from regression in a continuous space to classification in a discretized space. This is because classifying light directions into predefined bins of angles is much easier than directly regressing the unit vector itself. However, this conversion to inexact light directions obviously limits the learning of accurate surface normals. These methods have to strike a balance between learning difficulty and accuracy, which poses challenges for effectively estimating surface normals.

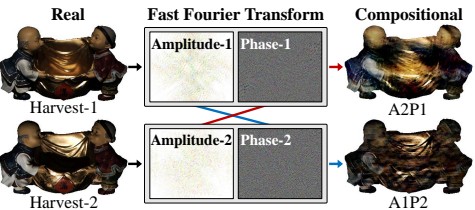

Figure 1: Motivation. We observed that shape information and light information can be "decomposed" in the Fourier domain. Amplitude and phase are generated by Fast Fourier Transform (FFT) and the compositional images are obtained by Inverse FFT (IFFT). Swapping the phase of two photometric stereo images with different light directions produces two compositional photometric stereo images with changed illuminated lights. Lines of the same color indicate a set of IFFT. Contrast is adjusted for easier viewing. In the name of the compositional images, A and P are abbreviations for amplitude and phase.

To overcome the aforementioned challenges, we propose a new framework that uses a one-stage Fourier Embedding network to handle UPS, without explicitly learning light directions. Our approach differs significantly from existing solutions that process images in the spatial domain. It is motivated by our observation of photometric stereo images in the Fourier domain: lighting information predominantly concentrates on phase, while shape information is closely related to amplitude. We analyze photometric stereo images in the Fourier domain and provide a concise illustration in Figure 1, where the shape and lighting can be "decomposed" into amplitude and phase in the Fourier domain, respectively. Swapping the phases of two photometric stereo images with different light directions produces two compositional photometric stereo images with changed illuminated lights (more discussion can be found in Section 3.1). This observation inspires the design of our framework, which handles the information of lighting and shape in a Fourier-embedded one-stage network without explicitly learning the light directions. Our method therefore can pay more attention to the shape information. This design has several advantages, as it avoids the difficulty of explicitly learning exact light directions in a two-stage network and physically decomposes lighting and shape information through Fourier transformation. Experimental results on synthetic and real datasets demonstrate the effectiveness of our method in addressing the UPS problem.

## 2 RELATED WORK

**Calibrated Photometric Stereo (CPS).** Classic photometric stereo Woodham (1980) assumes that only Lambertian (diffuse) reflectance exists on the surface of the target object, allowing shapes to be recovered using the least squares method. However, real-world objects only barely possess

the property of Lambertian reflectance. Traditional photometric stereo algorithms tackled non-Lambertian photometric effects through various approaches, including bidirectional reflectance distribution function (BRDF) modeling Ikehata & Aizawa (2014); Shi et al. (2014), outlier region rejection Wu et al. (2010); Ikehata et al. (2012), and exemplar-based techniques Hui & Sankaranarayanan (2016); Hertzmann & Seitz (2005). Readers can refer to Shi et al. (2019) for a comprehensive survey on these non-learning-based methods. In recent years, deep learning-based methods have been widely used in the context of photometric stereo Chen et al. (2018); Ikehata (2018); Li et al. (2019); Chen et al. (2020a); Honzátko et al. (2021); Ju et al. (2022b); Yao et al. (2020); Santo et al. (2017). Santo et al. (2017) were the pioneers in introducing a fully connected deep photometric stereo network for estimating pixel-wise surface normals, but it is limited to a fixed number of observations. To handle a variable number of observations, some subsequent works handle pixels into an observation map in a per-pixel manner Ikehata (2018); Li et al. (2019); Zheng et al. (2019), while others extract global cues from patches for normal estimation in an all-pixel manner Chen et al. (2018; 2020a); Ju et al. (2022b). Additionally, recent techniques Yao et al. (2020); Honzátko et al. (2021) combine both strategies to extract local and global features for more effective normal estimation. Details can be found in surveys by Zheng et al. (2020); Ju et al. (2022a). However, these works assume known lighting conditions and cannot effectively handle uncalibrated photometric stereo. Calibrating light sources can be a tedious process, which requires professional knowledge and may be unavailable in real-world applications. It is more convenient for the community if photometric stereo methods can operate without the need for ground-truth light directions.

**Uncalibrated Photometric Stereo (UPS).** UPS methods aim to automatically calibrate lighting conditions, eliminating the need for explicit knowledge of light directions. However, under the assumption of a Lambertian surface, solving UPS introduces GBR ambiguity, which is an inherent inability due to the lack of light source directions. To address this ambiguity, traditional works have been developed to provide additional knowledge, such as inter-reflections Chandraker et al. (2005), specular spikes Drbohlav & Chaniler (2005), parametric specular reflection Georghiades (2003), isotropic specular reflection Wu & Tan (2013), *etc*. With the recent advancements in neural networks, deep learning-based methods have achieved state-of-the-art performance in addressing the UPS problem. Chen et al. (2019; 2020b) proposed two-stage networks, which first estimate light conditions and then learn surface normals with both light information and images. Tiwari *et al.* jointly train the UPS network with image relighting and use multiple loss functions to optimize the network. Sarno *et al.* leveraged a differentiable neural architecture search (NAS) strategy to automatically find the most efficient neural architecture. Kaya et al. (2021) first used an uncalibrated neural inverse rendering approach to deal with unknown lights, and Li & Li (2022b) further allowed the re-rendered errors to be back-propagated to the light sources and refined them jointly with the normals.

However, all these learning-based UPS methods rely on explicit light direction estimation. As discussed, estimating light directions and then inputting them with photometric stereo images may lead to training instability and complicated training steps. In contrast, we propose a new framework that uses a one-stage Fourier Embedding network to handle UPS, without explicitly learning light directions. Our method avoids the difficulty of explicitly learning exact light directions in a two-stage network and physically decomposes lighting and shape information through Fourier transformation.

## 3 METHOD

### 3.1 MOTIVATION

Our main inspiration arises from observing the relationship between two photometric stereo images and the characteristics of their amplitude and phase components in the frequency domain through Fourier transform. As shown in Figure 1, when we swap the amplitudes of two images with different illumination, the resulting compositional images almost preserve the original shading cues. However, when we swap the phase of the input images in the Fourier domain, the compositional images also interchange the light directions. Therefore, we conclude that the light information and shape information can be decomposed to a certain extent, into the phase and amplitude in the Fourier domain, respectively. This inspires us to process light and shape cues separately in the Fourier domain.

Theoretically, the shape cues can be extracted solely from the amplitude spectrum. However, the swapped compositional images in Figure 1 are noisy and blurry, indicating that the "decomposition"

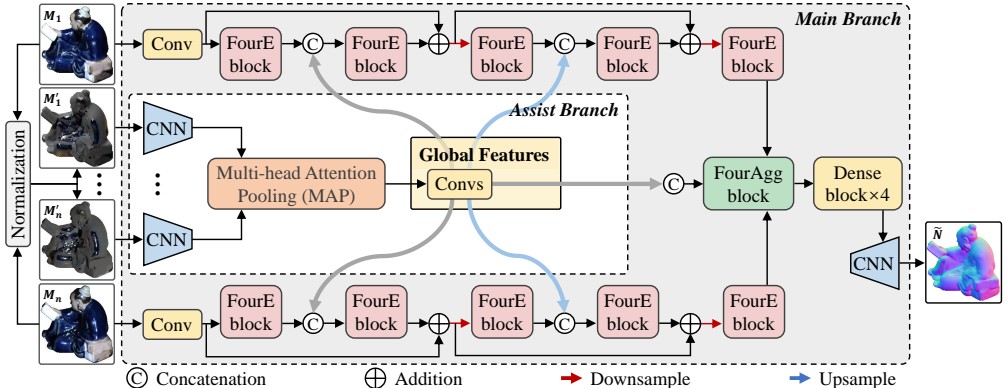

Figure 2: Overview of UPS-FourNet for surface normal estimation. Our approach can be viewed as a two-branch multi-input-single-output (MISO) network. The main branch takes the original images as input, which comprises five Fourier Embedding (FourE) blocks along with a Fourier Embedding Aggregation (FourAgg) block. Meanwhile, the assist branch takes the normalized images as input and passes them to an encoder network, which is partially based on the backbone structure of PS-FCN Chen et al. (2018) (shown as the blue trapezia). This branch is designed to couple with the main branch and assist in the learning process for surface normals with global features.

in the Fourier domain remains incomplete. Therefore, we also extract features from the phase spectrum to assist in learning surface normals. In conclusion, this observation allows us to develop a learning-based UPS method that no longer explicitly estimates light directions, which can alleviate the problems associated with the two-stage frameworks.

## 3.2 THE UPS-FOURNET

The two-branch networks have proven successful in numerous vision tasks Chen et al. (2021); Yu et al. (2022), which can be attributed to each branch focusing on its own information processing procedure. By effectively utilizing the distinct information from each branch and combining them appropriately, comprehensive information can be harnessed to significantly enhance the performance of surface normal estimation. Motivated by this idea, we propose our two-branch UPS-FourNet, as shown in Figure 2. Our UPS-FourNet is also a multi-input-single-output (MISO) network, because deep photometric stereo networks have to handle a variable number of input images.

The main branch of UPS-FourNet takes original photometric stereo images as inputs, while the assist branch feeds normalized images as inputs to assist in the learning process. Motivated by the observation in Section 3.1, we propose the Fourier Embedding (FourE) block and the Fourier Embedding Aggregation (FourAgg) block to handle features in the Fourier domain. We detail these two key components in Sections 3.3 and 3.4. Specifically, the main branch comprises five FourE blocks organized in a residual manner He et al. (2016), with two downsampling operations performed using bilinear interpolation. Additionally, one FourAgg block is used to handle a variable number of extracted features. Subsequently, we incorporate a 24-layer DenseNet module with four Dense blocks Huang et al. (2017), followed by the same structure of the regressor of PS-FCN Chen et al. (2018), to regress the estimated surface normals.

In the assist branch, we first adopt the normalization operation Chen et al. (2020a) to mitigate the impact of spatially-varying BRDFs. This is because the CNN-based framework operates on patch-level inputs and is trained with homogeneous BRDF. As shown in Figure 2, the extractor of the assist branch shares the same structure as the counterpart of PS-FCN Chen et al. (2018). However, the aggregation model that fuses a flexible number of features into one is different from the previous all-pixel-based photometric stereo networks Chen et al. (2018; 2020a); Ju et al. (2022b); Sarno et al. (2022). We introduce the Multi-head Attention Pooling (MAP) model Lee et al. (2019), drawing inspiration from its applications in Ikehata (2021; 2022). This model allows us to shrink the number of elements in the set from an arbitrary dimension $n$ to one by incorporating a learnable query $Q$, instead of solely retaining the maximum value as in Chen et al. (2018). The MAP model Lee et al. (2019) can be viewed as a global fusion method that considers all feature distributions for surface normal estimation, rather than only retaining the maximum value.

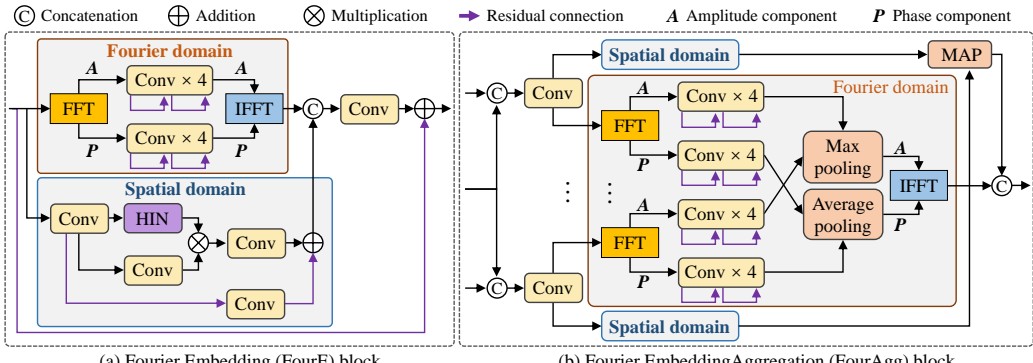

Figure 3: Structure of the proposed (a) Fourier Embedding (FourE) block, and (b) Fourier Embedding Aggregation (FourAgg) block.

Note that the FourE blocks in the main branch, along with the upsampling and $1 \times 1$ convolutional layer to adjust the spatial and channel dimensions, are concatenated with the features of the assist branch at different scales, as illustrated in Figure 2. This design enables the incorporation of global information in both the spatial and channel domains. On the one hand, the output of the assist branch keeps 1/4 of the original resolution, while the concatenated features in the main features are the same and half of the original resolution, respectively. Consequently, combining the output of the assist branch fuses features with different receptive fields, providing global information in the spatial domain. On the other hand, the output of the assist branch fuses features from all shading cues from different illumination directions, while the features in the main branch are extracted from a single photometric image. Therefore, combining the assist branch output and the main branch features integrates information from both local and global cues, enriching the channel domain. This approach enhances the network's capability to capture comprehensive information for surface normal estimation. We utilize different $1 \times 1$ convolutional layers (Convs in Fig .2) to adjust the channel for different concatenations.

### 3.3 FOURIER EMBEDDING BLOCK

In Section 3.1, we discovered that shape information and light information can be partially decomposed through Fourier transform. Therefore, we propose the Fourier Embedding (FourE) block to perform simultaneous feature extraction on amplitude and phase in the Fourier domain, along with feature enhancement in the spatial domain, inspired by Li et al. (2023). As shown in Figure 3 (a), the input features are split into the Fourier and Spatial domains. In the Fourier domain, Fast Fourier Transform (FFT) is employed to decompose the input into amplitude component ($A$) and phase component ($P$). These components then undergo four $3 \times 3$ convolutional layers, in the form of two residual blocks He et al. (2016). Finally, the features are transformed back to the spatial domain through Inverse Fast Fourier Transform (IFFT). In the Spatial domain, we enhance the features through an efficient Half Instance Normalization (HIN) model Chen et al. (2021) connected in parallel with a $3 \times 3$ convolutional layer. Note that the spatial domain is implemented with ResNet, so the input features are added to produce the final output.

Following feature extraction and enhancement in both the spatial and Fourier domains, we combine their outputs because they are complementary. We argue that spatial domain with convolutional layers can effectively model structural dependencies, while the Fourier domain can attend to global information and facilitate the disentanglement of shape and light. To further improve feature representation, we employ skip connections (element-wise addition) to combine the input feature and the combined feature, creating a residual structure He et al. (2016).

### 3.4 FOURIER EMBEDDING AGGREGATION BLOCK

As mentioned in Section 3.2, UPS-FourNet is a MISO network because photometric stereo needs to handle a variable number of input images. To cope with this variability, an additional fusion model is required to aggregate variable features into a representation with a fixed number of channels. This is necessary because CNN-based networks are not inherently equipped to handle a variable number

of inputs during training and testing Chen et al. (2020a). To address this limitation, we further propose the Fourier Embedding Aggregation (FourAgg) block to output aggregated features with a fixed number of channels for backpropagation.

As shown in Figure 3 (b), the FourAgg block shares a similar basic structure to the FourE block. After acquiring the extracted features from decomposed $A$ and $P$, we adopt different aggregation strategies for them. Specifically, max pooling is used to aggregate amplitude features, representing the shape cues extracted from $M_1$ to $M_n$. It emphasizes amplitude features, which avoid the impact of shadows and enhance fine details, thereby improving the representation of shape information. In contrast, we employ average pooling to aggregate phase features, representing the light cues extracted from $M_1$ to $M_n$. It is chosen to weaken the impact of phase features, resulting in a smoother representation of light information to help mitigate the ambiguity of light directions during surface normal estimation.

In the FourAgg block, we also incorporate the spatial domain to complement the Fourier domain. The spatial domain shares the same structure in the FourE block. However, different from the aggregation methods used in the Fourier domain, we use the MAP model Lee et al. (2019) (as described in Section 3.2) to aggregate spatial features, because it contains more comprehensive information on features from different light directions. Additionally, in the FourAgg block, each main branch feature is concatenated with the assist branch global feature (with a $1 \times 1$ convolutional layer to adjust the channel dimension) before the aggregation step. This design aims to further fuse global information in both the spatial and channel domains (as discussed in Section 3.2) and prevent information loss during aggregation operations. These strategies in the FourAgg block enable for effective feature fusion and enhance the performance of surface normal estimation.

## 3.5 LEARNING PROCEDURES

During training, we optimize the proposed UPS-FourNet by minimizing the following loss function $\mathcal{L}$, as follows:

$$\mathcal{L} = \|1 - \tilde{\boldsymbol{N}} \odot \boldsymbol{N}\|_1 + 0.1 \times \|\text{VGG}(\tilde{\boldsymbol{N}}) - \text{VGG}(\boldsymbol{N})\|_2, \tag{1}$$

where $\boldsymbol{N}$ is the ground truth and $\tilde{\boldsymbol{N}}$ is the estimated surface normals. The first term is the commonly used cosine similarity loss, and the symbol $\odot$ represents the element-wise product operation. If the estimated surface normals $\tilde{\boldsymbol{N}}$ have a similar orientation to the ground truth $\boldsymbol{N}$, $\tilde{\boldsymbol{N}} \odot \boldsymbol{N}$ will be close to 1, and the first term will tend to 0. In the second term, we also add a perceptual loss to enhance high-frequency details Johnson et al. (2016), with the weight factor empirically set to 0.1. The perceptual loss is computed using the pre-trained VGG-19 network, which is supervised at four scales.

Our network was implemented using PyTorch. The Adam optimizer is used with the default settings ($\beta_1 = 0.9$ and $\beta_2 = 0.999$) on a single RTX 3080 GPU. The initial learning rate is set to 0.002, divided by 2 every 5 epochs. We trained UPS-FourNet using a batch size of 32, for 40 epochs. The number of input images used for training is 32. In addition, the size of the input images during training is set to $32 \times 32$ pixels. Note that the number of input images and their size can be flexibly adjusted in testing. Our network was trained on the publicly synthetic Blobby and Sculpture shape datasets Johnson & Adelson (2011). For these shapes, we utilize the rendered photometric stereo images provided by Chen et al. (2019). The dataset comprises a total of 85,212 samples, each sample consisting of 64 images captured from different light directions sampled from the upper hemisphere. Among these images, 99%, *i.e.*, a total of 84,362 samples, were used to train our UPS-FourNet model. The remaining 852 samples were utilized for the purpose of validation.

## 4 EXPERIMENTS

To verify the quantitative accuracy of the estimated surface normals, we use the mean angular error (MAE) in degrees, calculated by MAE $= \frac{1}{U} \sum_p^U \cos^{-1}(\tilde{\boldsymbol{n}}^p \cdot \boldsymbol{n}^p)$, where $U$ is the total number of pixels in the area where the surface normals are considered, and $\tilde{\boldsymbol{n}}_{\boldsymbol{p}}$ and $\boldsymbol{n}_{\boldsymbol{p}}$ are the surface-normal vector at pixel $p$ of the ground-truth $\tilde{\boldsymbol{N}}$ and the estimated surface normals $\boldsymbol{N}$, respectively.

## 4.1 ABLATION STUDIES

We conducted several ablation studies to analyze the effectiveness of the main components of our design. Table 1 presents the quantitative comparison of the ablated models on the validation set. We report the average MAE across 852 samples, each with 64 input images. For the FourE block, we remove the Fourier domain (#1), remove the spatial domain (#2), and remove the HIN model Chen et al. (2021) in the spatial domain (#3). For the FourAgg block, we remove the spatial domain aggregation (#4), remove the Fourier domain aggregation (#5), and remove the concatenation of global features extracted from the assist branch (different from -GF in #12). Furthermore, we tested the aggregation methods for amplitude and phase features in Fourier domain aggregation, using average pooling for amplitude features and max pooling for phase features (inverted pooling methods) (#7), max pooling for all (#8), and average pooling for all (#9). For branches, we remove the assist branch (#10), remove the main branch (entails the removal of both the FourE block and FourAgg block) (#11), and remove the concatenation of global features to the main branch. Finally, we denote our complete model as #13.

Table 1: Quantitative comparison of ablation studies on our UPS-FourNet, in terms of average MAE, on the validation set, where FD: Fourier domain, SD: Spatial domain, FDA: Fourier domain aggregation, SDA: Spatial domain aggregation, GF: Global features, MB: Main branch, AB: Assist branch, -HIN: Without HIN model, ⇄P: Switched pooling methods for amplitude and phase aggregation, MP: All max pooling for aggregation, AP: All average pooling for aggregation, and -GF: Without global features.

| # | FourE Block | | FourAgg Block | | | Branches | | MAE (°) |
|---|---|---|---|---|---|---|---|---|
| | FD | SD | FDA | SDA | GF | MB | AB | |
| 1 | ✓ | | ✓ | ✓ | ✓ | ✓ | ✓ | 5.93 |
| 2 | | ✓ | ✓ | ✓ | ✓ | ✓ | ✓ | 6.04 |
| 3 | ✓ | -HIN | ✓ | ✓ | ✓ | ✓ | ✓ | 5.84 |
| 4 | ✓ | ✓ | ✓ | | ✓ | ✓ | ✓ | 6.11 |
| 5 | ✓ | ✓ | | ✓ | ✓ | ✓ | ✓ | 6.05 |
| 6 | ✓ | ✓ | ✓ | ✓ | | ✓ | ✓ | 5.91 |
| 7 | ✓ | ✓ | ⇄P | ✓ | ✓ | ✓ | ✓ | 6.39 |
| 8 | ✓ | ✓ | MP | ✓ | ✓ | ✓ | ✓ | 5.90 |
| 9 | ✓ | ✓ | AP | ✓ | ✓ | ✓ | ✓ | 6.05 |
| 10 | ✓ | ✓ | ✓ | ✓ | | ✓ | | 6.36 |
| 11 | | | | | | | ✓ | 6.92 |
| 12 | ✓ | ✓ | ✓ | ✓ | ✓ | -GF | ✓ | 5.88 |
| 13 | ✓ | ✓ | ✓ | ✓ | ✓ | ✓ | ✓ | 5.75 |

As shown in Table 1, we can see that all the key designs contribute to the optimal performance achieved by the full model. The absence of the Fourier domain (#2 and #5) results in a significant drop in MAE, signifying the pivotal role of decomposing amplitude and phase in the Fourier domain to enhance surface normal estimation involving unknown lighting. From the results of #2 and #4, we demonstrate that the spatial domain can also boost the accuracy of surface normal estimation, showing the complementary information it contains. Meanwhile, ablation #3 shows the effectiveness with the use of the utilized HIN model Chen et al. (2021). Moreover, we tested the performance of different aggregation methods (#7, #8, and #9) in the FourAgg block. Compared to our default settings, using two different pooling methods (#7) to fuse phase and amplitude features leads to a significant performance decrease in comparison to the results under our default settings. This trend echoes in #8 and #9, which shows the necessity of emphasizing shadow features in the amplitude component, while reducing the influence of light in the phase component. Finally, we also verified that the two-branch structure is beneficial to surface normal estimation (#10), and concatenating global information (#12) enhances accuracy. Note that when we exclude the main branch with FourE and FourAgg blocks (#11), the model degrades to the original single-stage method similar to UPS-FCN Chen et al. (2018), which cannot handle UPS well.

## 4.2 EVALUATION ON BENCHMARKS AND REAL DATASETS

We first evaluate our UPS-FourNet and compare it with previous calibrated and uncalibrated methods on the widely used photometric stereo dataset, namely the DiLiGenT benchmark Shi et al. (2019). DiLiGenT contains 10 objects and each object has 96 images captured under different lighting conditions. The quantitative results for surface normal estimation are tabulated in Table 2. We compare UPS-FourNet with recent state-of-the-art calibrated and uncalibrated learning-based photometric stereo methods. Additionally, Figure 4 provides visual representations of the reconstruction results and error map comparisons for the "Reading" and "Harvest" objects. Our method produces superior results for regions with specular highlights and cast shadows.

To conduct a comprehensive analysis of the generalization capability of our UPS-FourNet across various objects and materials, we further use the challenging DiLiGenT10$^2$ dataset Ren et al. (2022). DiLiGenT10$^2$ further contains 100 objects of 10 shapes multiplied by 10 materials and each object has 100 images under different conditions. These datasets pose significant challenges due to their inclusion of strongly non-Lambertian surface materials and complex structures. The results, as obtained from the online evaluation website, are presented in Figure 5.

Table 2: Performance on the DiLiGenT benchmark Shi et al. (2019) with 96 images, in terms of MAE (degrees). UPS-1s stands for the one-stage methods without learning light information, while UPS-2s stands for the two-stage methods that take the first-stage estimated light directions and intensities as the input of the second-stage surface normal network.

| Method | Task | Ball | Bear | Buddha | Cat | Cow | Goblet | Harvest | Pot1 | Pot2 | Reading | Avg. |
|---|---|---|---|---|---|---|---|---|---|---|---|---|
| IRPS Taniai & Maehara (2018) | CPS | 1.47 | 5.79 | 10.36 | 5.44 | 6.32 | 11.47 | 22.59 | 6.09 | 7.76 | 11.03 | 8.83 |
| PS-FCN Chen et al. (2018) | CPS | 2.82 | 7.55 | 7.91 | 6.16 | 7.33 | 8.60 | 15.85 | 7.13 | 7.25 | 13.33 | 8.39 |
| GPS-Net Yao et al. (2020) | CPS | 2.92 | 5.07 | 7.77 | 5.42 | 6.14 | 9.00 | 15.14 | 6.04 | 7.01 | 13.58 | 7.81 |
| CNN-PS Ikehata (2018) | CPS | 2.12 | 8.30 | 8.07 | 4.38 | 7.92 | 7.42 | 14.08 | 5.37 | 6.38 | 12.12 | 7.62 |
| NA-PSN Ju et al. (2022b) | CPS | 2.93 | 5.48 | 7.12 | 4.65 | 5.99 | 7.49 | 12.28 | 5.96 | 6.42 | 9.93 | 6.83 |
| LL22a Li & Li (2022a) | CPS | 2.43 | 3.64 | 8.04 | 4.86 | 4.72 | 6.68 | 14.90 | 5.99 | 4.97 | 8.75 | 6.50 |
| PX-NetLogothetis et al. (2021) | CPS | 2.03 | 4.13 | 7.61 | 4.39 | 4.69 | 6.90 | 13.10 | 5.08 | 5.10 | 10.26 | 6.33 |
| UPS-FCN Chen et al. (2018) | UPS-1s | 6.62 | 11.23 | 15.87 | 14.68 | 11.91 | 20.72 | 27.79 | 13.98 | 14.19 | 23.26 | 16.02 |
| KS21 Kaya et al. (2021) | UPS-2s | 3.78 | 5.96 | 13.14 | 7.91 | 10.85 | 11.94 | 25.49 | 8.75 | 10.17 | 18.22 | 11.62 |
| SDPS-Net Chen et al. (2019) | UPS-2s | 2.77 | 6.89 | 8.97 | 8.06 | 8.48 | 11.91 | 17.43 | 8.14 | 7.50 | 14.90 | 9.51 |
| SK22 Sarno et al. (2022) | UPS-2s | 3.46 | 5.48 | 10.00 | 8.94 | 6.04 | 9.78 | 17.97 | 7.76 | 7.10 | 15.02 | 9.15 |
| UPS-GCNet Chen et al. (2020b) | UPS-2s | 2.50 | 5.60 | 8.60 | 7.80 | 8.48 | 9.60 | 16.20 | 7.20 | 7.10 | 14.90 | 8.70 |
| LERPS Tiwari & Raman (2022) | UPS-2s | 2.41 | 6.93 | 8.84 | 7.43 | 6.36 | 8.78 | 11.57 | 8.32 | 7.01 | 11.51 | 7.92 |
| LL22b Li & Li (2022b) | UPS-2s | 1.24 | 3.82 | 9.28 | 4.72 | 5.53 | 7.12 | 14.96 | 6.73 | 6.50 | 10.54 | 7.05 |
| UPS-FourNet (Ours) | UPS-1s | 2.49 | 5.62 | 7.45 | 4.85 | 6.14 | 7.79 | 13.91 | 6.10 | 6.66 | 11.42 | 7.24 |

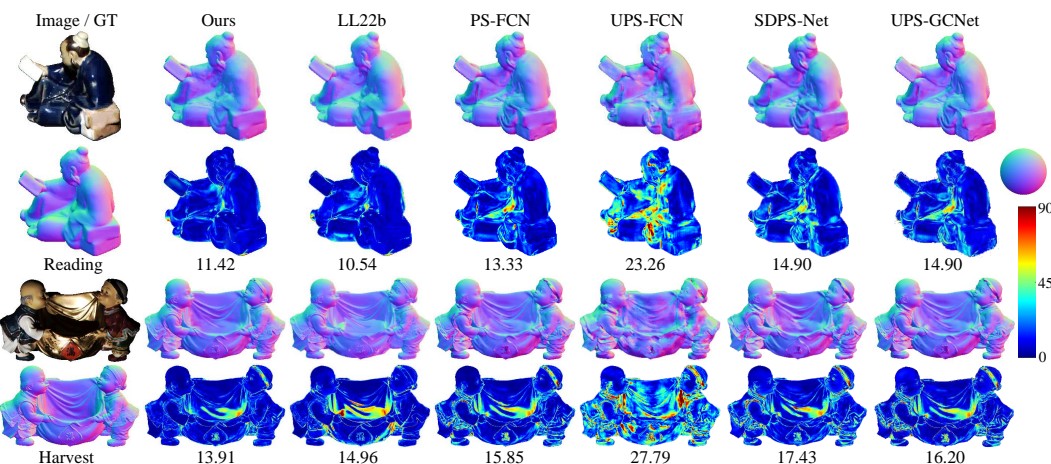

Figure 4: Quantitative results on the DiLiGenT dataset with 96 input images are presented. In each sample, the first row displays the estimated normal maps, while the second row depicts the error maps obtained from various methods. The values indicate the MAE in degrees. The contrast of the images has been adjusted to improve visualization.

Finally, we evaluate our method using the more intricate Light Stage Data Gallery dataset Einarsson et al. (2006), which incorporates general non-Lambertian materials, complex structures, and lower-quality images. As ground truth data is unavailable for this dataset, we present qualitative results for the "Helmet" and "Plant" objects using our method in Figure 6. These results encompass surface normals and 3D reconstruction outcomes obtained via Simchony et al. (1990), utilizing 32 randomly selected input images from a pool of 253 images. It can be seen that the reconstruction of details is evident via our method.

### 4.2.1 DISCUSSION

As shown in Table 2, our UPS-FourNet achieves the second-best performance among other state-of-the-art UPS methods. Note that our method belongs to the one-stage UPS method, without explicitly learning the light information. Compared to the previous one-stage method UPS-FCN Chen et al. (2018) (also shown in Figure 5), UPS-FourNet obviously improves the estimation of surface normals, which is attributed to the decomposition of shape and lighting. Our one-stage framework has several advantages over previous two-stage UPS methods. Two-stage methods tend to require longer training times and tedious training operations because they involve learning lighting features separately and then feeding them into another network for surface normal estimation. Additionally, the two-stage UPS approaches may suffer from unstable optimization. The instability arises from inaccurate initial lighting estimation, which may affect surface learning and lead to local op-

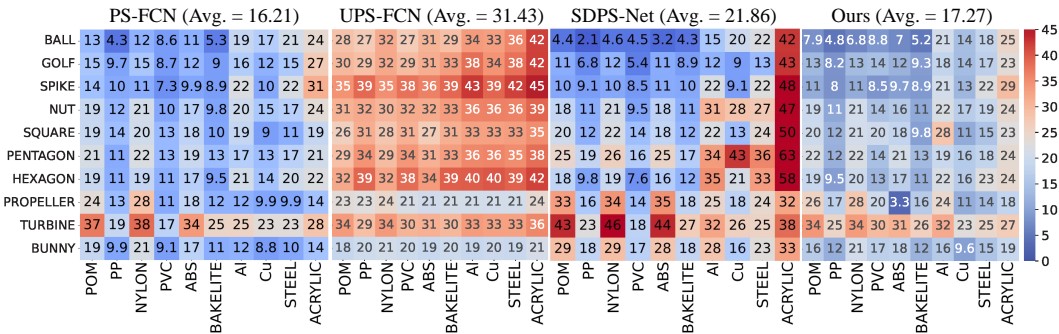

Figure 5: The shape-material error matrix used to compare our UPS-FourNet with recent calibrated and uncalibrated methods. The number in each element of the matrix represents the MAE in degrees according to the shape and material index.

timal learning. On the other hand, it introduces further instability in surface learning through the conversion of light direction estimation from continuous regression to discrete classification.

In fact, UPS-FourNet outperforms other UPS methods on objects with complex structures and strong non-Lambertian surfaces in the DiLiGenT dataset, as shown in Table 2. The visual results in Figure 4 illustrate the superior performance of our method in the regions with specular reflections ("Reading") and cast shadows ("Harvest"). However, on some very simple objects, such as "Ball" and "Bear", the recent two-stage method LL2b Li & Li (2022b) obviously shows more reasonable results, because these objects are easy to acquire supervision by inverse rendering. Conversely, the simple structure may lead to inefficient feature extraction in the decomposed amplitude and phase components of UPS-FourNet.

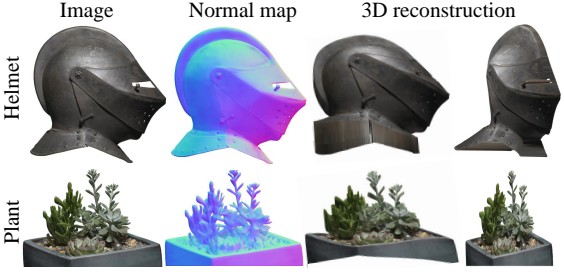

Figure 6: Evaluation on the Light Stage Data Gallery, with 32 input images. The estimated surface normals are shown qualitatively. The 3D reconstruction results of our estimated surface normal maps are illustrated using Simchony et al. (1990). The contrast of the images is adjusted for easier visualization.

## 5 CONCLUSION

In this paper, we propose a Fourier transform-based one-stage UPS method, namely UPS-FourNet. Our approach is motivated by the unique characteristics observed when swapping the phases of two photometric stereo images captured under different lighting directions, where the shape and light information can be "decomposed" in the Fourier domain. Consequently, our method eliminates the need for explicit lighting learning, distinguishing it from the two-stage UPS methods. It relaxes the limitations of previous two-stage UPS methods, with better training stability, concise end-to-end structures, and avoiding discrete classification errors in estimating the light directions. Ablation studies highlight the effectiveness of the proposed modules and experimental results on widely used benchmarks demonstrate the competitive performance of UPS-FourNet. Specifically, we significantly improve the accuracy of the one-stage pipeline and our method achieves the second-best results among all the UPS methods.

**Limitations and future work:** Currently, UPS-FourNet normalizes the intensity of photometric stereo images during training, *i.e.*, intensity calibration is not considered. This aspect will be a focus of our future work. Additionally, although UPS-FourNet shows better performance on complex structures and strongly non-Lambertian surfaces, it does not achieve the best results in terms of average MAE on the DiLiGenT benchmark. Note that UPS-FourNet represents an initial exploration of the one-stage learning-based UPS framework without incorporating many advanced modules or structures, *e.g.*, Transformer. Our future work will delve into the development of more accurate one-stage UPS models.

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
