# OpenReview forum: "Light-Implicit Uncalibrated Photometric Stereo Network With Fourier Embedding"
_ICLR.cc/2024/Conference — ICLR 2024 Conference Withdrawn Submission_

### Official Review · Reviewer_hxLh · 2023-10-26

**Soundness:** 2 fair
**Presentation:** 2 fair
**Contribution:** 2 fair
**Rating:** 3
**Confidence:** 5

**Summary:**

Well-known uncalibrated deep photometric stereo works generally rely on the two-stage pipeline, i.e., in the first stage, they predict light source, and in the second stage, they infer surface normals. On the contrary, this paper proposes a one-stage deep neural network pipeline resorting to the well-known Fourier transform of images. The paper further claims that the amplitude and phase of the transformed images can provide useful cues to estimate light, thereby avoiding the need to learn light sources in the photometric stereo setup explicitly. Experiments on conventional benchmark datasets are conducted to show the suitability of the proposed pipeline.

**Strengths:**

* An attempt to overcome the two-stage approach to uncalibrated photometric stereo problem.
* Using the ideas from signal processing domain to a classical computer vision problem is compelling.

**Weaknesses:**

* The authors do not provide a theoretical backup on the phase and amplitude swap argument presented in the paper.
* Since the paper is a one-shot, the paper should have conducted experiments showing that the current method could also help resolve GBR. No such experiments or theories are presented in the paper.
* No training time comparison, memory footprint, or model complexity comparisons are provided in the paper.
* Despite paper claims to be suitable for non-Lambertian objects,  it is not justified why such an approach is more suitable for non-Lambertian objects.
* Performance is inferior to previous art, not a major weakness, but a weakness nonetheless.
* Paper is not well-written. Furthermore, visual presentation must be improved, for instance Fig. 1 is too small to visualize the presented idea. On equations, the dimension of the introduced variables is missing.

Refer Questions section for detailed comments.

**Questions:**

## Abstract
* “fewer interpretations” -> I request the authors to clarify what kind of interpretations we are talking about.
* two-stage methods discretize the light direction estimations instead of regressing exact light directions due to the learning difficulty and instability. -> In fact the lights placed in the photometric stereo experimental setup are indeed discrete in nature. So I don’t see a problem with previous methods in making such a choice. Furthemore, I don’t see discrete light modeling could lead to instability. Please comment. Also, I request to kindly be specific on the statements made in the paper.
* “decomposing inputs” -> what is input here. Please be specific.
* Inconsistent use of “surface normals” and “shape” throughout. It's better to be consistent.  Go either with surface normal or shape. I recommend using surface normals.



## Introduction
* under multiple images -> using multiple images
* with different lights  -> lights are generally of the similar intensity level. So, not really different in nature.Yet, with many such LEDs/light sources placed at a distance from the object.  Please be on-point.
* Compared with geometric stereo methods -> Firstly, add citations. Secondly, I believe it should be “geometric multi-view stereo” methods.
* high-frequency details on objects’ surfaces -> add citations.
* owing to the powerful capabilities of deep neural net -> what capabilities of deep neural network?
* inputting -> using. It is good to be formal.

General Comment on Introduction:
Following up on the claims made in the abstract related to the non-Lambertian objects, refer to the very first line of the abstract. It is not justified in the introduction as to why such a method is more suitable for non-Lambertian objects. I don’t see any arguments placed by the authors in this regard. From the introduction what is clear to me is that the authors are proposing a method to suitably estimate surface normal without a need to explicitly learn lights using PS images. Kindly comment and refine the introduction accordingly.


## Method
* “Theoretically, the shape cues can be extracted solely from the amplitude spectrum”. Please add citations. I am really looking forward to a proof of this. Furthermore, I am interested in a test that could verify the argument presented in the paper for inter-reflecting symmetric objects like vase, refer Kaya et al. 2021 dataset. Moreover, I would like to see a theoretical backup on the phase and amplitude swap argument presented in paper. To be mindful, the product —normal.light ($N^T \cdot L$), the standard PS equation, will turn to convolutions in the fourier domain, referring to the modulation property of the fourier transform.

* “We argue that spatial domain with convolutional layers can effectively model structural dependencies, while the Fourier domain can attend to global information and facilitate the disentanglement of shape and light”. I am not completely satisfied by the later part of this statement. Fourier transform indeed can have global information, but does that facilitate disentanglement? Amplitude and phase of light has also to do with light intensity, occlusion, interreflection, polarization, etc. I further see a clear relation to the spherical harmonics theory in estimating normals from images referring to Basri and Jacobs TPAMI 2003, Ramamoorthi and Hanrahan 2001. Unfortunately, these works are not discussed in this paper.

## Evaluation and Benchmark
* The performance is clearly inferior to the previous art in uncalibrated PS. Also, the paper must conduct experiments on the Kaya et al. 2021 dataset, which to me is more challenging to validate the paper’s claims.
* Since the paper is one-shot, they should have conducted the experiments showing that the current method could also help resolve GBR. No such experiments or theories are presented in the paper.
* No train time and test time comparison, memory footprint and model complexity comparisons are provided in the paper.

**Details Of Ethics Concerns:**

The paper appears to have no ethical concerns.

---

### Official Review · Reviewer_xXVv · 2023-10-27

**Soundness:** 3 good
**Presentation:** 3 good
**Contribution:** 2 fair
**Rating:** 5
**Confidence:** 5

**Summary:**

This paper introduces a one-stage deep uncalibrated photometric stereo network, named UPS-FourNet, for non-Lambertian objects.The authors first present their observation that light and shape information can be decomposed to a certain extent into the phase and amplitude in the Fourier domain. Based on this observarion,  they propose the Fourier Embedding (FourE) block to extract features simultaneously from the amplitude and phase in the Fourier domain along with feature enhancement in the spatial domain, and the Fourier Embedding Aggregation (FourAgg) block to aggregate the amplitude and the phase features, respectively, from multiple photometric stereo images for direct normal regression without explicitly estimating light directions. Experiments show competitive performance.

**Strengths:**

- The obsevation that light and shape information can be partially decomposed into phase and amplitude in the Fourier domain sounds novel. This can be helpful in the design of UPS methods.
- The proposed method is end-to-end and can regress normal directly without explicit light estimation.
- Aggregating amplitude and phase features separately from multiple photometric stereo images sounds novel.
- Experimental results show improvements over existing one-stage UPS methods, which are also comparable to two-stage methods.
- Ablation study has been carried out to evaluate the effectiveness of the core components in the proposed method.

**Weaknesses:**

- The idea of embedding Fourier transform into deep networks is not new. The FourE block basically has the same architecture as the FouSpa block in UHDFour [Li et al. 2023], and the FourAgg block simply performs separate amplitude and phase features aggregation which is a simple logical choice for feature fusion in the Fourier domain.
- The proposed one-stage method only focuses on normal regression and cannot recover light information. It is also not clear whether or not the input photometric images have been normalized by the input light intensities.
- Some claims are not well supported such as the decomposition of the shape and light and the role of Fourier/spatial domain.
- Other than performing feature extraction in the Fourier domain, the proposed design does not seem to have fully exploit the observation presented by the authors.
- The evaluations are not comprehensive and detailed enough.

[Li et al. 2023] Chongyi Li, Chun-Le Guo, Zhexin Liang, Shangchen Zhou, Ruicheng Feng, Chen Change Loy, et al. Embedding fourier for ultra-high-definition low-light image enhancement. In ICLR, 2023.

**Questions:**

- The claim in Section 3.1 is not well supported. From Figure 1 it can be observed that the shading or lighting variance information is included in phase spectrum. However, it cannot demonstrate the shape information is encoded in amplitude spectrum, as the shapes of the two images are the same. The authors of [Li et al. 2023] also performed the same analysis and arrived at a slightly different conclusion. Are their conclusion compitable with the one drawn in this paper? If not, what leads to the differences?
- In Section 3.3, it said “We argue that spatial domain with convolutional layers can effectively model structural dependencies, while the Fourier domain can attend to global information and facilitate the disentanglement of shape and light.” However, there’s no analysis to support this claim. The authors should try to analyze their roles with visualizations.
- The actual effects of the submodules are not clear enough and not analyzed in detail. It’s better to visualize the sequential feature maps in both Fourier and spatial domain to better support the claims.
- What’s the training and testing time compared to the existing one/two-stage UPS methods?
- The ablation study is incomplete. For instance, what’s the performance of “w/o normalization”, “w/o dense block”, “w/o MAP”, etc.?
- What if only amplitude information or only phase information is considered in the framework?
- What’s the performance of the proposed method with different number of input images in test time? What is the minimium number of images for getting a plausible prediction?

---

### Official Review · Reviewer_pDmL · 2023-10-31

**Soundness:** 1 poor
**Presentation:** 2 fair
**Contribution:** 2 fair
**Rating:** 3
**Confidence:** 5

**Summary:**

This paper introduces a Fourier transform-based approach to the uncalibrated photometric stereo problem. This method bypasses the estimation of light direction and directly estimates the surface normal, referred to as a "one-stage" process. A central contention of this paper is that the proposed UPS-FourNet implicitly discerns lighting by decomposing inputs using an embedded Fourier transform.
Authors claim that the lighting information is primarily concentrated in phases, while shape information correlates strongly with amplitudes. Subsequently, a neural network is designed to learn from these features, achieving results that are competitive with prior works.

**Strengths:**

The concept of processing amplitudes and phases in two distinct branches and merging them at the end is innovative. However, the paper contains several uncertainties and unsupported claims.

**Weaknesses:**

One-stage vs. Two-stages: I respectfully disagree with one of the primary assertions in the paper that the "one-stage" UPS method is superior to the "two-stage" UPS method. The two-stage UPS methods referenced in this paper treat light direction estimation as a byproduct, which can be advantageous for many downstream tasks.

I agree with that most two-stage methods suffer from inaccurate light estimation, which can consequently introduce errors into the normal estimation stage.
However, some recent works, such as LL22b, have managed to simultaneously optimize the light direction, resulting in enhanced normal/light estimation comparable in accuracy to calibrated photometric stereo. Thus, I am of the opinion that simultaneous estimation of light direction and normal isn't necessarily a drawback.

The paper's assertion—that "lighting information predominantly concentrates on phases while shape information is closely related to amplitudes of the Fourier domain"—lacks sufficient evidence. From Fig 1, it's not evident whether phases indeed yield more lighting information than amplitudes. To further substantiate this claim, I recommend that the authors conduct an ablation study: discard the amplitude branch and observe the affected regions of the object, then do the same for phases. The quantitative assessment in Tab1 is insufficient; additional visualizations would aid readers in gaining a clearer understanding.

The selection of max pooling and average pooling in the FourAgg block lacks proper validation. The quantitative assessment in Tab1 is insufficient to substantiate the authors' claim in Sec-3.4, at the top of page 6, which states: "Max pooling emphasizes amplitude features, ... thus enhancing the representation of shape information. ... Average pooling is employed to diminish the influence of phase features, ... aiding in reducing the ambiguity of light directions." More visual evidence is needed to validate this assertion. For instance, the authors could illustrate which features are captured by each pooling operation and how these learned features influence the final results in specific regions.

There appears to be a typo error at the beginning of page 7, where it reads, "For the FourE block, we remove the Fourier domain (#1), and remove the spatial domain (#2)," when Tab 1 indicates the opposite.

**Questions:**

In summary, many of the claims in this paper are not well substantiated, and much of its insights appear to be more empirical than theoretical, making them potentially less generalizable to subsequent studies or works in other areas. Consequently, I lean towards not accepting this paper.

---

### Official Review · Reviewer_rCYN · 2023-10-31

**Soundness:** 2 fair
**Presentation:** 3 good
**Contribution:** 2 fair
**Rating:** 3
**Confidence:** 5

**Summary:**

This paper proposes a one-stage uncalibrated photometric stereo method based on Fourier transform. Two modules are designed to accomplish this task, namely, Fourier Embedding block which extracts features from Fourier domain and spatial domain and Fourier embeddingaggregation block which is designed to fuse features extracted from multiple inputs.

**Strengths:**

The observation that shape and lighting can be decomposed through Fourier transform is interesting and worth further exploring.

**Weaknesses:**

Though the observation is interesting, it lacks enough theoretical and experimental proof to make it a solid contribution.

The proposed method also can’t achieve SOTA results on DiLiGenT dataset when compared to 2-stage UPS methods.

**Questions:**

Though the observation seems to be interesting and somewhat promising. It still needs more proof, either theoretical or experimental, to understand why and how much Fourier transform can help the decomposition of shape and lighting cue.

Also, Since the results is good but not best, the claimed advantages over 2-stage UPS methods are weak according to my opinion. So given the current state of the work, it’s premature to be published on ICLR.